# Investigation of Mechanical and Tribological Behaviors of Aluminum Based Hybrid Metal Matrix Composite and Multi-Objective Optimization

**DOI:** 10.3390/ma15165607

**Published:** 2022-08-16

**Authors:** Biranu Kumsa Gonfa, Devendra Sinha, Umesh Kumar Vates, Irfan Anjum Badruddin, Mohamed Hussien, Sarfaraz Kamangar, Gyanendra Kumar Singh, Gulam Mohammed Sayeed Ahmed, Nand Jee Kanu, Nazia Hossain

**Affiliations:** 1Program of Mechanical Design and Manufacturing Engineering, Department of Mechanical Engineering, School of Mechanical, Chemical and Materials Engineering, ASTU, Adama 1888, Ethiopia or; 2Amity School of Engineering and Technology, Amity University Uttar Pradesh, Noida 201301, India; 3Mechanical Engineering Department, College of Engineering, King Khalid University, Abha 61421, Saudi Arabia,; 4Department of Chemistry, Faculty of Science, King Khalid University, Abha 61413, Saudi Arabia; 5Pesticide Formulation Department, Central Agricultural Pesticide Laboratory, Agricultural Research Center, Dokki, Giza 12618, Egypt; 6Center of Excellence (COE) for Advanced Manufacturing Engineering, Program of Mechanical Design and Manufacturing Engineering, School of Mechanical, Chemical and Materials Engineering, ASTU, Adama 1888, Ethiopia; 7Department of Mechanical Engineering, JSPM Narhe Technical Campus, Pune 411041, India; 8School of Engineering, RMIT University, Melbourne, VIC 3001, Australia

**Keywords:** metal matrix composites, GRA, Al/SiC/MoS_2_, Al 6061, silicon carbide, stirring time, stirring speed, multi-response optimization

## Abstract

Aluminum metal matrix composites are potential materials for aerospace and automobile industrial applications due to their enhanced mechanical and tribological properties. Aluminum reinforced with silicon carbide particles has been developed with enhanced mechanical and tribological behavior, but it lacks wettability between matrix and reinforcement causing weak bonding, which reduces the degree of enhancement. The objectives of this study were to fabricate aluminum-based metal matrix composites with enhanced wettability at varying stirring speeds (350, 450, 550 rpm), stirring time (5, 10, 15 min), weight percentage of SiC (0, 5, 10 wt.%), and weight percentage of MoS_2_ (0, 2, 4 wt.%). Nine samples were fabricated using stir casting based on Taguchi L9 orthogonal array. Hardness, tensile strength, and wear rate of the developed composite were investigated and analyzed as a single response characteristic using Taguchi’s signal-to-noise ratio and as a multi-response characteristic using hybrid Taguchi–grey relational analysis (HTGRA). The results revealed that the addition of SiC in the composite produced better hardness, tensile strength, and wear rate. The addition of MoS_2_ in the composite showed better hardness and tensile strength only up to 2 wt.% of MoS_2_, and in the case of wear rate, the addition of MoS_2_ in the composite up to 4% showed better wear resistance. Al–SiC–MoS_2_ hybrid composite shows better enhancement in hardness, tensile strength, and wear resistance than the Al–SiC composite.

## 1. Introduction

Aerospace and automobile industries require a lightweight, harder, stronger, stiffer, and wear resistant material [1,2]. In nature, no material satisfies this requirement; a composite is the only material that fulfills this industrial requirement [2,3]. Composite material combines two or more dissimilar materials to create a material with better behavior than either of the originals alone [4,5]. The combination of materials must have two main phases, a matrix phase and a reinforcing phase [6,7]. There are three basic types of composites based on the matrix phase: metal matrix composite, ceramic matrix composite, and polymer matrix composite [8,9]. Metal matrix composite (MMC) has hard particles for reinforcement and metal matrix to enhance the mechanical and tribological behavior of the composite. The matrix material has to be lightweight, have high strength-to-weight ratio, and high modulus-to-weight ratio. The materials that satisfy this behavior are aluminum, titanium, and magnesium [1,2,7].

Metal matrix composites, therefore, can be aluminum alloy-based composites [9,10], magnesium alloy-based composites [6,11], or titanium alloy-based composites [4]. The materials used as reinforcement in the production of MMCs are aluminum oxide (Al_2_O_3_), silicon carbide (SiC), titanium bromide (TiB_2_), titanium carbide (TiC), titanium nitride (TiN), and boron carbide (B_4_C) [12,13]. Of all MMCs, aluminum alloy-based composites have shown better improvement in the mechanical and wear properties needed by industries [14,15]. Aluminum alloys are termed versatile materials to be used for numerous engineering applications because of their better machining, joining, and processing properties [16,17]. In addition, the low cost, increased strength-to-weight ratio, and other environmentally friendly characteristics of aluminum alloys make them preferable materials in engineering applications [11,14,15]. There are several series of aluminum alloys (e.g., 1xxx, 2xxx, 5xxx, 6xxx, 7xxx, and others) used in the production of composites. Among aluminum alloys, the 6xxx and 7xxx series have good machinability and extrudability and are mostly used as a matrix material in the production of AMMC aerospace industry, architectural construction, marine industries, and automotive applications [3,18,19]. 

Taguchi method is a technique for designing and performing experiments to investigate processes where the output depends on many factors. Taguchi uses the signal-to-noise (S/N) ratio to measure the quality of characteristics deviating from the desired value. Three categories of signal-to-noise ratios (larger is better, smaller is better, and the nominal is best) are used to find the optimal parameters, and the signal-to-noise ratio of each parameter level must be assessed for each output function [19,20]. Grey relational analysis is widely used for optimizing multi-response parameters by converting multi-responses into a single response. This method has been applied by several researchers to optimize the control parameters having multi-responses through grey relational grade.

## 2. Materials and Methods

### 2.1. Materials

Al 6061 is used in this paper due to its corrosion resistance and moderate strength, and its composition is tabulated in Table 1. Al 6061 is extensively used because of its low weight, low cost, and good formability and weldability. Silicon carbide particle (53 µm) reinforcement was used to develop this composite, and it certainly improves the overall strength of the composite. The processing parameters to manufacture the desired samples are given in Table 2. Aluminum MMC reinforced with SiC particles has an improvement in mechanical behavior compared to unreinforced aluminum matrix alloy [12]. Molybdenum disulfide (MoS_2_) is used as a dry or solid lubricant to improve the poor wettability that occurs between Al 6061 and SiC because of its low coefficient of friction [9]. Table 3 depicts the experimental parameters and their levels. Table 4 represents the experimental design for composite fabrication.

### 2.2. Experimental Process Parameters and Design of Experiments

Fabrication of the composite was achieved based on the basic input variables of process parameters (stirring speed, stirring time, and weight percentage of SiC and MoS_2_) at the same pouring temperature (730 °C) for each fabrication. 

Taguchi L9 orthogonal array experimental design was used for fabrication based on the given factors and levels. In the design of the experiment table for fabrication, the letters A, B, C, and D are used to represent the process parameters of stirring speed, stirring time, wt.% of SiC, and wt.% of MoS_2_, respectively, to indicate the optimization of S/N ratio plot.

### 2.3. Conducted Testing

Hardness testing was conducted as per ASTM using the Vickers hardness tester machine shown in Figure 1a and the sample shown in Figure 1d. The sample used for hardness testing was a cylindrical form with a size of 20 mm in length and 20 mm in diameter. Tensile strength testing was conducted based on the ASTM E8 standard using Shimadzu (Kyoto, Japan) AG-X plus TM 50 kN universal testing machine as shown in Figure 1b and the sample shown in Figure 1e. The sample size has gauge length (G = 43.75 mm), R = 6.25 mm, overall length (L = 82.5 mm), parallel length (A = 50 mm), and grip width (C = 14 mm) as shown in Figure 1g. Wear rate testing was conducted based on ASTM G99-05 using pin-on-disc (DUCOM TR-20 MICRO) equipment as shown in Figure 1c and the sample shown in Figure 1f. The sample used for wear testing was a cylindrical form with a size of 12 mm in length and 6 mm in diameter.

### 2.4. Analysis of Particle Distribution in Matrix

The microstructural image was analyzed by determining the grain particle size area from the scaled dimension of each grain particle using ImageJ software. ImageJ software considers and counts all grain particles in the Al matrix of the composite and determines the volume fraction of the particles in the Al matrix. The data recorded from this software were the total cross-sectional area of the analyzed images, the total number of grain particles, the total area of the grain particles, the average area of grain particle size, and grain particle volume fraction in the matrix. In the analysis, there are two phases, as indicated by red and white in Figure 2. The red color indicates the total area of grain particles in the matrix obtained from the total area of the analyzed image.

From the analysis, the volume fractions of grain particles were proportional to each other’s (i.e., the higher reinforcement weight of contents in the matrix, the higher the volume fraction). Sample 3, which has the higher weight percentage of SiC and MoS_2_, shows the higher volume fraction; Sample 1, which has no weight percentage of SiC and MoS_2_, shows the smaller volume fraction in the microstructural analysis as shown in Table 5.

## 3. Results 

Table 6 shows the experimental results of hardness, tensile strength, and wear rate of the developed composite.

The hardness test was performed using a Vickers hardness tester machine. From Table 6, the highest and lowest values of hardness were 208 HV and 123 HV, respectively. When the wt.% of SiC increases (at a higher level) and wt.% of MoS_2_ is at 2 wt.%, the hardness of the composite increases. In contrast, the lowest hardness was recorded as the wt.% of SiC, and wt.% of MoS_2_ is lower (at a lower level). The hardness of the composite increased due to increasing content of SiC particles and strong bonding between SiC particles and matrix due to enhanced wettability with the addition of MoS_2_. SiC particles increase the grain boundaries in composites and the grain size of the composites are reduced. Dislocation mobility will be restricted or challenged to move from grain to grain due to increased grain boundaries (decreased grain size) since grain boundaries act as a barrier to the motion of dislocation. Moreover, SiC particles have an extremely low coefficient of thermal expansion and how they strain the atomic lattice of the aluminum matrix, resulting in a dramatic increase in dislocation density. Dislocation density is the average distance between dislocation decreases and the dislocation starts blocking the motion of each other. The addition of MoS_2_ also shows an enhancement in hardness up to 2 wt.% and shows a decrement with 4 wt.%, which means the role of MoS_2_ is only facilitating the wettability, so only a small amount is enough to enhance the wettability and bonding between both matrix and reinforcement to each other, and further increment leads to decreasing the hardness of the composite.

The tensile strength testing was performed using the universal testing machine (UTM). The largest values of tensile strength were indicated in sample number seven and the lowest value in sample number one with values of 194 MPa and 75 MPa, respectively. When the wt.% of SiC increases (at the higher level) and wt.% of MoS_2_ is at 2 wt.%, the tensile strength of the composite increases. In contrast, the lowest tensile strength was recorded as the wt.% of SiC, and wt.% of MoS_2_ is lower (at the lower level). The tensile strength of the composite is increased due to increasing contents of SiC particles and strong bonding between SiC particles and matrix due to enhanced wettability with the addition of MoS_2_. SiC particles increase the grain boundaries in composites and the grain size of the composites are reduced. Dislocation mobility will be restricted or challenged to move from grain to grain due to increased grain boundaries (decreased grain size) since grain boundaries act as a barrier to the motion of dislocation. 

Moreover, SiC particles have an extremely low coefficient of thermal expansion and how they strain the atomic lattice of the aluminum matrix, resulting in a dramatic increase in dislocation density. The addition of MoS_2_ also shows an enhancement in tensile strength up to 2 wt.% and shows a decrement with 4 wt.%, which means the role of MoS_2_ is only facilitating the wettability, so only a small amount is enough to enhance the wettability and bonding of both matrix and reinforcement to each other and further increment leads to decrease the tensile strength of the composite.

The wear rate test was performed using pin-on-disc apparatus testing machine. The wear rate study was conducted based on the mass loss analysis. The largest of the wear rate values was indicated in sample number three and the lowest value was in sample number one, with the values of 10 × 10^−9^ kg/m and 1 × 10^−9^ kg/m, respectively. When the wt.% of SiC increased (at the higher level) and wt.% of MoS_2_ increased (higher level), the wear rate of the composite decreased. In contrast, the highest wear rate was recorded as the wt.% of SiC, and wt.% of MoS_2_ is lower (at the lower level). The wear resistance of the composite is increased due to the increasing content of SiC particles and MoS_2_ reinforcement because the wear resistance of carbides is very high, and the hard-reinforcing particles work to resist wear on the surface of the cast hybrid composite samples. Another reason for the improved wear resistance of the cast AMC is the adequate interfacial bonding between the hybrid reinforcement and the aluminum matrix due to the addition of MoS_2_, which resists the pull-out of the hybrid reinforcement during the relative movement between two contacting surfaces. The addition of MoS_2_ reinforcement in Al 6061/SiC composites as a hybrid reinforcement further increases the wear and friction resistance of the composite.

## 4. Discussion

### 4.1. Analysis of Hardness

In the figure, by increasing the weight percentage of SiC, the hardness of the developed composite increased at constant wt.% MoS_2_ (0%). At the higher wt.% of SiC, the higher value of hardness is shown. Silicon carbide has the most significant effect on the hardness of the developed composite’s mechanical properties such as tensile strength, hardness, and impact strength, but the high amount of SiC will lead to brittleness [7]. From Figure 3, the hardness of composites increased with increasing wt.% of SiC and the hardness values for 0, 5, and 10 wt.% SiC at constant wt.% MoS_2_ (0%) are 123 HV, 180 HV, and 199 HV, respectively. This is a better result than the previous studies by [1,2].

MoS_2_ is used as a solid lubricant because it does not increase the hardness of composites, but it facilitates the wettability between aluminum and silicon carbide due to its low friction properties and robustness to enhance the hardness result. The hardness is better when a small amount of MoS_2_ is added to Al–SiC composite. The hardness of the matrix increases as the weight percentage of MoS_2_ increases, but only up to 2%, and after then it declines at 4 wt.% MoS_2_ at constant weight percentage of SiC. The values of hardness for 0, 2, and 4 wt.% MoS_2_ at constant weight percentage of SiC (0%) are 123 HV, 180 HV, and 163 HV, respectively, as shown in Figure 3. The highest hardness value of the developed composite was shown when the weight percentage of SiC was 10% and the weight percentage of MoS_2_ was 2%, and this shows a better result than the previous studies by [2]. Al–SiC–MoS_2_ hybrid composite shows better enhancement in hardness than Al–SiC composite.

#### Optimization of Process Parameters for Hardness

In the response to hardness variation for data analysis and prediction of optimum results, Taguchi signal-to-noise ratios were used. In this study, the effects of varying control factors (stirring speed, stirring time, wt.% of SiC, and wt.% of MoS_2_) on the responses of hardness were analyzed. As higher values of hardness were desirable, larger, better-quality characteristic was selected to investigate the influence of factors on hardness response. 

The delta statistics of the S/N ratio for the hardness tabulated in Table 7 show the ranks of the factors affecting the hardness response based on the S/N ratios. Ranks were assigned based on their delta value. The delta values were calculated from the difference between the largest and smallest value of the mean value. The analysis showed that wt.% SiC was assigned a rank of 1 with a delta value of 2.48, signifying that it is the predominant factor that affects the hardness of the composite. The wt.% MoS_2_, stirring time, and stirring speed were assigned second (1.26), third (1.15), and fourth (0.88) ranks, respectively.

Figure 4 shows the parametric effect S/N ratio plot and the optimal parameter combination for the higher hardness. The numerical value of the maximum point in each graph shows the best optimum combination of the factors at that level. Therefore, the S/N ratio plot in Figure 4 shows the maximum point in each graph is at (A) stirring speed at level 2, (B) stirring time at level 3, (C) wt.% SiC at level 3, and (D) wt.% MoS_2_ at level 2, with the corresponding values of 450 rpm, 15 min, 10% wt.% SiC, and 2% wt.% MoS_2_, respectively (i.e., **A2B3C3D2**). The optimum prediction condition for the S/N ratio in which the higher hardness result of the composite obtained with the term setting of **A2B3C3D2** is **229 HV**. Wt.% of SiC has the greatest impact on the hardness of the developed composites and is followed by wt.% of MoS_2_ (second), stirring time (third), and stirring speed (fourth).

### 4.2. Analysis of Tensile Strength

In the figure, when increasing the wt.% of SiC, the tensile strength of the developed composite increased at constant wt.% MoS_2_ (0%). At the higher weight percentage of SiC, the higher value of tensile strength is shown. Silicon carbide has the most significant effect on the tensile strength of the fabricated composite’s mechanical properties such as tensile strength, hardness, and impact strength, but the high amount of SiC will lead to brittleness [7]. 

From Figure 5, the tensile strength of composites increased with increasing the weight percentage of SiC and the tensile strength values for 0, 5, and 10 wt% SiC at constant wt.% MoS_2_ (0%) are 75 MPa, 119 MPa, and 181 MPa, respectively. This result shows a better result than the previous studies by [1,18]. 

MoS_2_ is used as a solid lubricant because it does not increase the hardness of composites, but it facilitates the wettability between aluminum and silicon carbide due to its low friction properties and robustness to enhance the tensile strength. As a result, the tensile strength is better when a small amount of MoS_2_ is added to Al-SiC composite. 

Tensile strength of the matrix increases as the weight percentage of wt.% MoS_2_ increases, but only up to 2%, and after then it declines at 4 wt.% MoS_2_. The values of tensile strength for 0, 2, and 4 wt.% MoS_2_ at constant weight percentage of SiC (0 wt.%) are 75 MPa, 90 MPa, and 82 MPa, respectively, as shown in Figure 5. The highest tensile strength value of the developed composite was shown when the weight percentage of SiC was 10% wt. of SiC and the weight percentage of MoS_2_ is 2% and this result shows a better result of tensile strength than the previous studies by [2]. Al–SiC–MoS_2_ hybrid composite shows better enhancement in tensile strength than Al–SiC composite.

#### Optimization of Process Parameter for Tensile Strength

In the response of hardness variation for data analysis and prediction of optimum results, Taguchi signal-to-noise ratios were used. In this study, the effects of varying control factors (stirring speed, stirring time, wt.% of SiC, and wt.% of MoS_2_) on the responses of tensile strength were analyzed.

The delta statistics of the S/N ratio for the tensile strength tabulated in Table 8 show the ranks of the factors affecting tensile strength responses based on the S/N ratios. Ranks were then assigned based on their delta value. The delta values were calculated from the difference between the largest and smallest value of the mean values. The analysis showed that wt.% SiC was assigned a rank of 1 with a delta value of 6.90, signifying that it is the predominant factor that affects the tensile strength of the composite. Wt.% MoS_2_, stirring time, and stirring speed were assigned second (1.23), third (0.67), and fourth (0.40) ranks, respectively.

Figure 6 shows the parametric effect S/N ratio plot and the optimal parameter combination for the higher tensile strength. The numerical value of the maximum point in each graph shows the best optimum combination of the factors at that level. Therefore, the S/N ratio plot shows the maximum points in each graph are (A) stirring speed at level 3, (B) stirring time at level 2, (C) SiC at level 3, and (D) MoS_2_ at level 2, with the corresponding values of 550 rpm, 10 min, 10%, and 2% (i.e., **A3B2C3D2**). The optimum prediction condition for the S/N ratio in which the higher tensile strength results when the composite obtained with the term setting of **A3B2C3D2** is **201 MPa**. Wt.% of SiC has the greatest impact on the tensile strength of the developed composites followed by wt.% of MoS_2_ (second), stirring time (third), and stirring speed (fourth).

### 4.3. Analysis of Wear Rate

As shown in the figure, when increasing the weight percentage of SiC, the wear rate of the developed composite decreased. At the higher weight percentage of SiC, the smaller value of wear rate is shown. Silicon carbide has the most significant effect on the wear rate of the developed composites and since SiC is a wear-resistant material, the wear rate of the composite decreases due to the addition of SiC [15]. From Figure 7, the wear rate of the composite decreased with increasing wt.% of SiC and the wear rate values for 0, 5, and 10 wt.% SiC at constant wt.% MoS_2_ (0%) are 10, 8, and 4 ×10^−9^ kg/m, respectively. This result shows a better result than the previous studies by [17].

MoS_2_ used as solid lubricant and lubrication is one way to prevent wear. It facilitates the wettability between aluminum and silicon carbide due to its low friction properties and robustness to enhance the wear resistance of composite by decreasing the wear rate. The wear rate is smaller when the weight percentage of MoS_2_ is increased to Al–SiC composite. 

The wear rate of Al matrix decreases as the wt.% MoS_2_ increases. The values of wear rate for 0, 2, and 4 wt.% MoS_2_ at a constant weight percentage of SiC (0%) are 10, 9.002, and 9 × 10^−9^ kg/m, respectively, as shown in Figure 7. The smallest wear rate value of the developed composite was shown when the weight percentage of SiC was 10% wt. of SiC and the weight percentage of MoS_2_ was 4%. This shows a better result of wear rate than the previous studies by [2]. Al–SiC composite shows lower enhancement in decreasing wear rate than Al–SiC–MoS_2_ hybrid composite.

#### Optimization of Process Parameter for Wear Rate

The delta statistics of S/N ratio for the wear rate tabulated in Table 9 show the ranks of factors affecting wear rate response based on the S/N ratios. Ranks were then assigned based on their delta value. The delta values were calculated from the difference between the largest and smallest value of the mean values. The analysis showed that the wt.% SiC was assigned a rank of 1 with a delta value of 12.196, signifying that it is the predominant factor that affects the wear rate of the cast composite. The wt.% MoS_2_, stirring time, and stirring speed were assigned second (5.152), third (3.626), and fourth (3.263) ranks, respectively.

Figure 8 shows the parametric effect S/N ratio plot and the optimal parameter combination for the lower wear rate. The numerical value of the maximum point in each graph shows the best optimum combination of the factors at that level. Therefore, the S/N ratio plot shows the maximum point in each graph at (A) stirring speed at level 1, (B) stirring time at level 3, (C) wt.% SiC at level 3, and (D) wt.% MoS_2_ at level 3, with the corresponding values of 350 rpm, 15 min, 10% wt.% SiC, and 3% wt.% MoS_2_, respectively (i.e., **A1B3C3D3**). The optimum prediction condition for the S/N ratio in which the lower wear rate results in the composite is obtained when the term setting of (i.e., **A1B3C3D3**) is **1**. Wt.% of SiC has the greatest impact on the wear resistance of the developed composites followed by wt.% of MoS_2_ (second), stirring time (third), and stirring speed (fourth).

### 4.4. Multi-response Optimization

For multi-response optimization purposes, hybrid Taguchi with grey relational analysis (HTGRA) was used for stirring speed, stirring time, wt.% SiC, and wt.% MoS_2_ process parameters. 

Steps taken during optimization using GRA:


**Step 1: Transformation of data into S/N ratios.**


The S/N ratio of the experimental results of hardness, tensile strength, and wear rate of the developed composite were generated with the help of Minitab 17 software and are tabulated in Table 10.


**Step 2: Normalization of S/N values.**


Normalization of S/N values is a generation of grey relational and normalized data sequences for the experimental results within 0 and 1. The equations used were Equation (1) for “larger is better”, i.e., for hardness and tensile strength, and Equation (2) for “smaller is better”, i.e., for the wear rate S/N ratio response [19].
*Zij* = Normalized value 
(1)$$Zij=\frac{Yij-\min\left(Yij,\;i=1,2,3,\ldots .n\right)}{\max\left(Yij,\;i=1,2,3,\ldots .n\right)-\min\left(Yij,\;i=1,2,3,\ldots .n\right)}$$

(For larger is better, i.e., for hardness and tensile strength)
(2)$$Zij=\frac{\min\left(Yij,\;i=1,2,3,\ldots .n\right)-Yij}{\max\left(Yij,\;i=1,2,3,\ldots .n\right)-\min\left(Yij,\;i=1,2,3,\ldots .n\right)}$$

(For smaller is better, i.e., for wear rate)

Equation (1) is used for normalizing the value of S/N ratio for hardness and tensile strength, and Equation (2) is used for normalizing the value of S/N ratio for wear rate. Max *Yij* and Min *Yij* for hardness are 46.3738 and 41.8193, respectively. Max *Yij* and Min *Yij* for tensile strength are 45.7753 and 37.5012, respectively. Max *Yij* and Min *Yij* for wear rate are 0.0000 and −20.0000, respectively (Table 11). 


**Step 3: Determination of deviation sequence and grey relational coefficient (*GRC*).**

(3)
$$GRCij=\frac{\Delta \min+\partial \Delta \max}{\left(\Delta ij+\partial \Delta \max\right)}$$

where *i* = 9 (number of experiments) and *j* = 3 (number of responses)*GRCij* = GRC for the *i*th experiment/trial and *j*th dependent variable/responseDeviation sequence (Δ)= (max of normalized values–corresponding normalized value) *Yoj* = optimum performance value or the ideal normalized value of the *j*th response*Yij* = the *i*th normalized value of the *j*th response/dependent variableΔmin = smallest value of Δ and Δmax = highest value of  Δ∂ is the distinguishing coefficient (0 ≤∂≤ 1) 


Equation (3) is used for determining the grey relational coefficient. Minimum (Δmin) and maximum deviation sequence (Δmax) are used. Δmin for hardness, tensile strength, and wear rate are 0, 0, and 0, respectively. ΔMax for hardness, tensile strength, and wear rate are 1, 1, and 1, respectively (Table 12).

**Step 4: Calculation of grey relational grade (*GRG*) and its order of sequencing**(4)$$GRGi=\frac{1}{m}\sum GRCi\;$$
where *m* (3 in this case) is the number of responses (hardness, tensile strength, and wear rate) (Table 13)

### 4.5. Optimization of Process Parameters Using GRA

#### Analysis of S/N Ratio for GRG

For the analysis of GRG, the larger is better signal-to-noise ratio has been used.

The delta statistics of the S/N ratio for the GRG tabulated in Table 14 show the ranks of the factors affecting the multi-responses based on the S/N ratios. Ranks were then assigned based on their delta value. The delta values were calculated from the difference between the largest and smallest value of the mean values. The analysis showed that wt.% SiC was assigned a rank of 1 with a delta value of 2.354, signifying that it is the predominant factor that affects the GRG response. Wt.% of MoS_2_, stirring speed, and stirring time were assigned second (1.624), third (1.110), and fourth (0.142) ranks, respectively.

Figure 9 shows the parametric effect S/N ratio plot and the optimal parameter combination for the higher GRG responses. The numerical value of the maximum point in each graph shows the optimum combination of the factors at that level. Therefore, the S/N ratio plot shows the maximum point in each graph is (A) stirring speed at level 3, (B) stirring time at level 2, (C) wt.% SiC at level 3, and (D) wt.% MoS_2_ at level 2, with the corresponding values of 550 rpm, 10 min, 10 wt.% SiC, and 2 wt.% MoS_2_, respectively (i.e., **A3B2C3D2**) was selected. The most effective parameter of GRG response is wt.% SiC when compared with other factors and stirring time has the least effect on the GRG. The optimum prediction condition for the S/N ratio in which the higher (multi-response characteristics) GRG result of the composite is obtained with the terms set at (A) stirring speed at level 3, (B) stirring time at level 2, (C) wt.% SiC at level 3, and (D) wt.% MoS_2_ at level 2, with the corresponding values of 550 rpm, 10 min, 10 wt.% SiC, and 2 wt.% MoS_2_, respectively (i.e., **A3B2C3D2**) (i.e., **A3B2C3D2**) are **0.923**.

## 5. Conclusions

In this experimental study, aluminum-based MMC at varying stirring speeds (350, 450, 550 rpm), stirring time (5, 10, 15 min), weight % of silicon carbide powder (0, 5, 10 wt.%), and weight % of MoS_2_ powder (0, 2, 4 wt.%) were prepared using stir casting. Microstructure, hardness, tensile strength, and wear behavior of the developed composites were studied. Based on the results, the following conclusions are drawn: 

The analysis of hardness, tensile strength, and wear resistance were performed with the help of the Taguchi S/N ratio for single response optimization and hybrid Taguchi–grey relational analysis for multi-response optimization. Optical micrographs showed homogenous dispersion of particles in the matrix. Porosities were found and it is higher for reinforcement contents are higher.

From the S/N ratio analysis, the addition of SiC in the composite showed better hardness, tensile strength, and wear resistance. Wt.% of SiC is the only and the most significant factor affecting the hardness, tensile strength of the composite, followed by wt.% of MoS_2_, stirring time, and stirring speed. In the case of wear resistance, only wt.% of SiC and wt.% of MoS_2_ are the significant factors and wt.% of SiC is the most significant factor affecting the wear rate, followed by wt.% of MoS_2_, stirring time, and stirring speed. Addition of MoS_2_ in the composite showed better hardness and tensile strength only up to 2 wt.% of MoS_2_ and in case of wear rate the addition of MoS_2_ in the composite up to 4% showed better wear resistance than unreinforced matrix.

Therefore, the maximum hardness = 208.30 HV has been obtained at stirring speed 450 rpm, stirring time 15 min, 10% wt.% of SiC particles, and 2% wt.% of MoS_2_, maximum tensile strength = 194.43 MPa has been obtained at stirring speed 550 rpm, stirring time 10 min, 10% wt.% of SiC particles and 2% wt.% of MoS_2_ and the lowest wear rate = 1 × 10^−9^ kg/m has been obtained at stirring speed 350 rpm, stirring time 15 min, 10 wt.% of SiC particles and 4 wt.% of MoS_2_. The optimum prediction condition for the higher hardness, tensile strength, and lowest wear rate has been obtained at A2B3C3D2, A3B2C3D2, and A1B3C3D3, respectively. From the grey relational analysis for multi-response characteristics, the optimum prediction condition for the S/N ratio has been obtained at stirring speed 550 rpm, stirring time 10 min, 10% weight fraction of SiC particles, and 2% weight fraction of MoS_2_ (i.e., A3B2C3D2). Al–SiC–MoS_2_ hybrid composite shows better enhancements in hardness, tensile strength, and wear resistance than the Al–SiC composite. Therefore, the enhancement of wettability has been achieved due to the addition of MoS_2_ in the Al–SiC composite.

## Figures and Tables

**Figure 1 materials-15-05607-f001:**
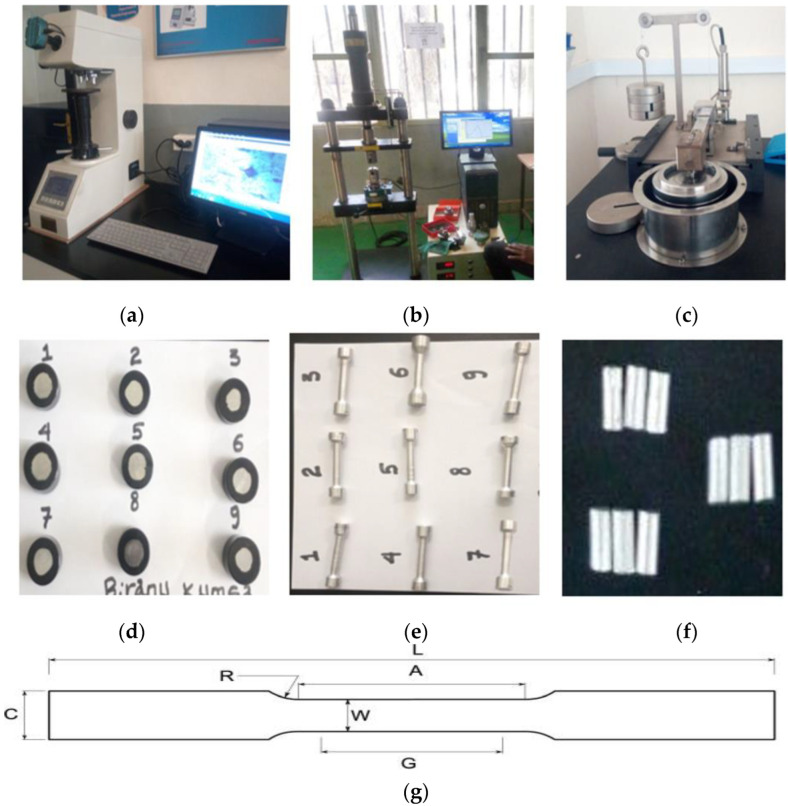
Testing devices and specimens. (**a**) Hardness testing; (**b**) Tensile testing; (**c**) Wear testing; (**d**) Hardness test sample; (**e**) Tensile test sample; (**f**) Wear test sample; (**g**) Sample size for tensile testing.

**Figure 2 materials-15-05607-f002:**
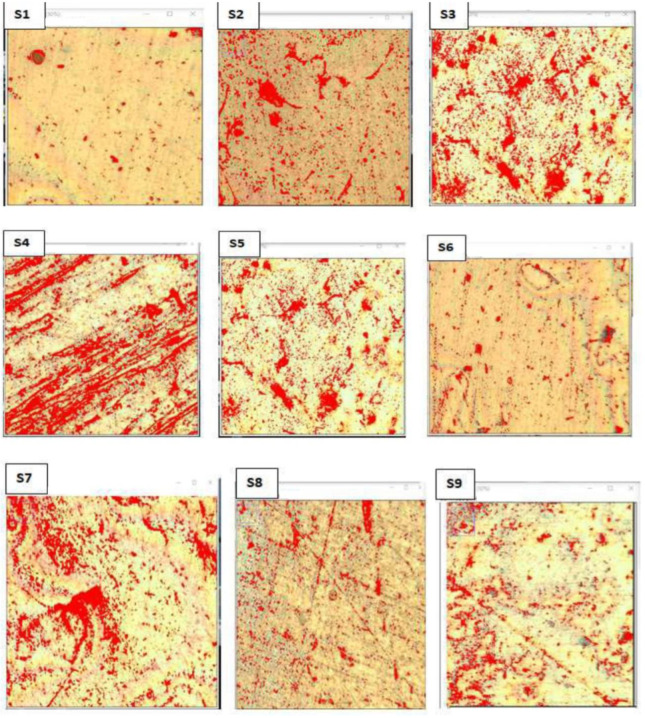
Optical microscopic images showing particle distribution.

**Figure 3 materials-15-05607-f003:**
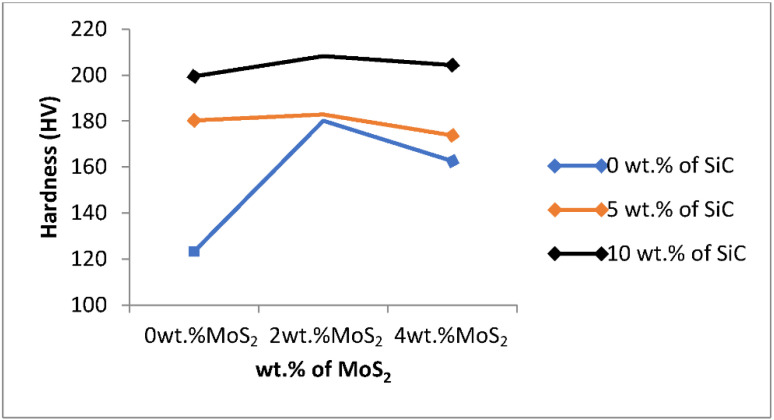
Effect of process parameters on hardness.

**Figure 4 materials-15-05607-f004:**
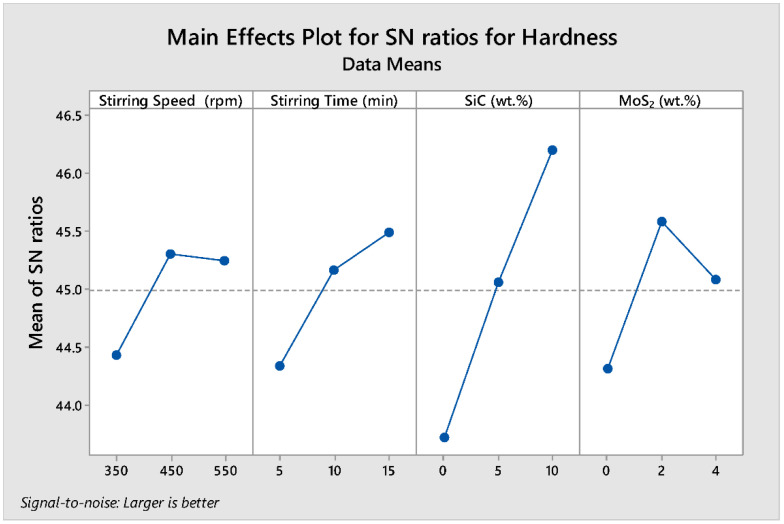
Main effect plots for S/N ratio of hardness.

**Figure 5 materials-15-05607-f005:**
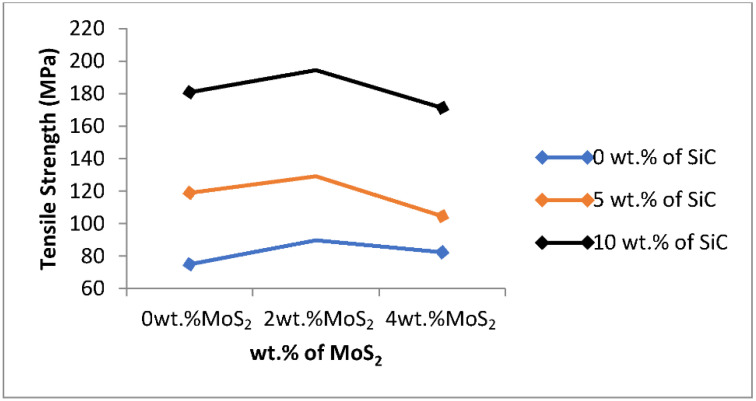
Effect of process parameters on tensile strength.

**Figure 6 materials-15-05607-f006:**
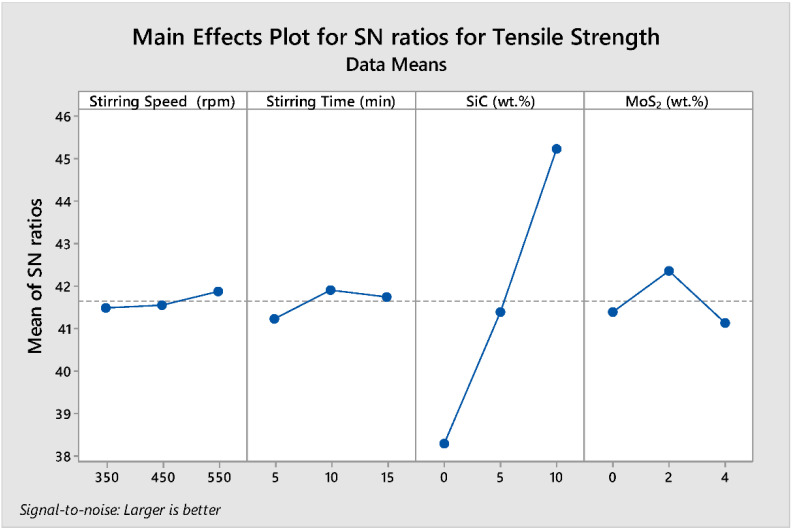
Main effect plots for S/N ratio of tensile strength.

**Figure 7 materials-15-05607-f007:**
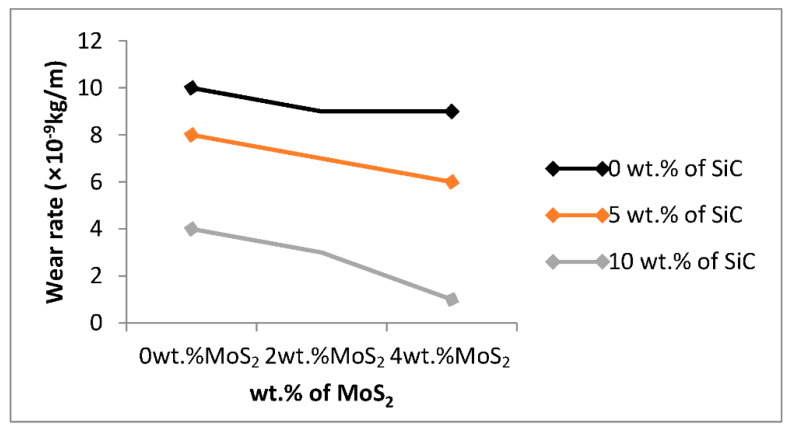
Effect of process parameters on wear rate.

**Figure 8 materials-15-05607-f008:**
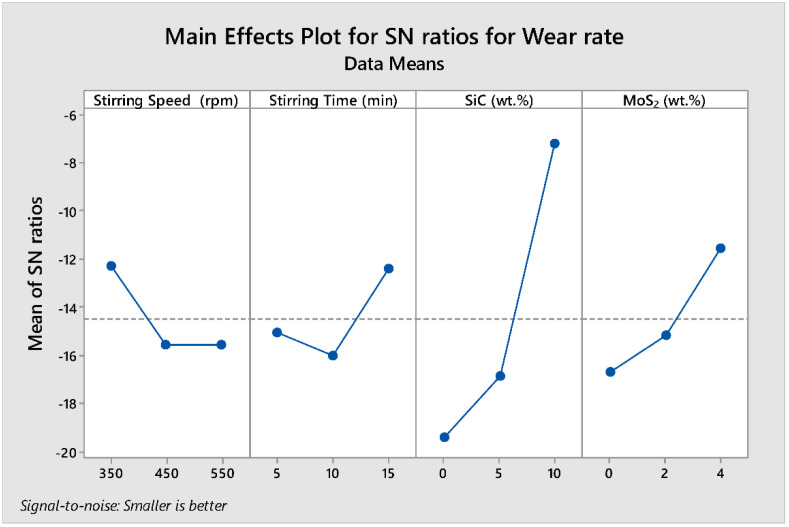
Main effect plots for S/N ratio of wear rate.

**Figure 9 materials-15-05607-f009:**
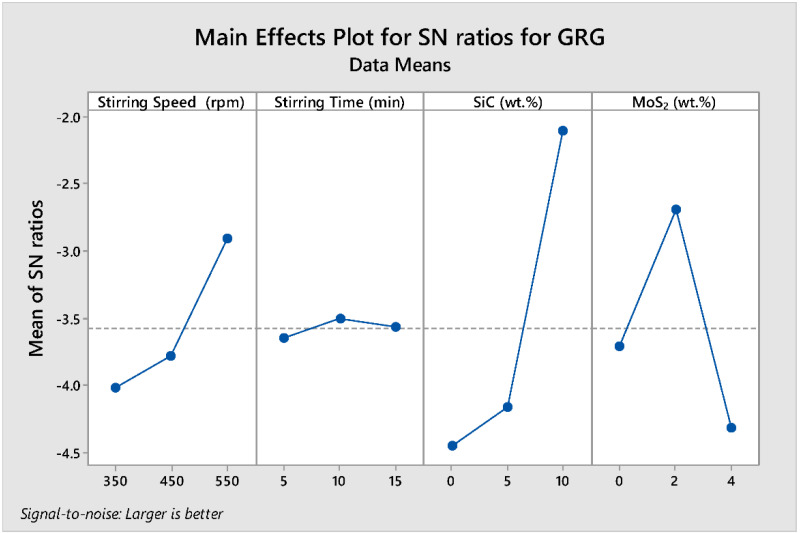
Main effect plots for S/N ratio of GRG.

**Table 1 materials-15-05607-t001:** Chemical weight composition of Al 6061.

**Elements of Al 6061 wt.%**	**Si**	**Mn**	**Mg**	**Cr**	**Zn**	**Cu**	**Fe**	**Ti**	**Al**
	**0.7**	**0.05**	**0.9**	**0.3**	**0.2**	**0.30**	**0.6**	**0.1**	**96.85**

**Table 2 materials-15-05607-t002:** Properties of Al 6061 and silicon carbide (SiC).

Property	Value	
**Formula**	Al 6061	SiC
Density	2.71 g/cm^3^	3.21 g/cm^3^
Melting point	605 °C	2730 °C
Modulus of elasticity	70 GPa	410 GPa
Tensile strength (σt)	115 MPa	3900 MPa
Poisson’s ratio (ν)	0.33	0.14
Form	Rod	Powder
Particle size	-	270 mesh (53 µm)
Hardness	30 Kg/mm^2^	2800 Kg/mm^2^

**Table 3 materials-15-05607-t003:** Experimental parameters and their levels.

S. No	Process Parameters	Levels			Units
**1**	**Stirring time (A)**	350	450	550	rpm
2	Stirring time (B)	5	10	15	min
3	wt.% of SiC (C)	0	5	10	%
4	wt.% of MoS_2_ (D)	0	2	4	%

**Table 4 materials-15-05607-t004:** Experimental design for composite fabrication.

S. No	A (rpm)	B (min)	C (wt.%)	D (wt.%)	Designation
1	350	5	0	0	S_1_
2	350	10	5	2	S_2_
3	350	15	10	4	S_3_
4	450	5	5	4	S_4_
5	450	10	10	0	S_5_
6	450	15	0	2	S_6_
7	550	5	10	2	S_7_
8	550	10	0	4	S_8_
9	550	15	5	0	S_9_

**Table 5 materials-15-05607-t005:** Determination of volume fraction of grain particles in the composite.

Designation	Total Area of Images (μm^2^)	Total Counted Grain Particles	Total Grain Particle Area (μm^2^)	Average Area of Grain Particle (μm^2^)	Volume Fraction Grain Particle Size (%)
S1	624,679.7	500	9264	18.528	1.483
S2	1261,693	1809	134,875	74.558	10.690
S3	1,970,589.4	2930	369,052	125.956	18.728
S4	1,360,391.2	998	155,071	155.382	11.399
S5	1,866,556.6	2551	248,700	97.491	13.324
S6	1,281,421.4	1762	56,075	31.825	4.376
S7	1,697,183.9	2091	239,863	114.712	14.133
S8	1,628,568.9	1930	95,369	49.414	5.856
S9	837,422.7	1846	75,837	41.082	9.056

**Table 6 materials-15-05607-t006:** Experimental results of hardness, tensile strength, and wear rate values.

Designation	Experimental Result		
	Hardness (HV)	Tensile Strength (MPa)	Wear Rate (×10^−9^ Kg/m)
S1	123	75	10
S2	183	129	7
S3	204	171	1
S4	174	105	6
S5	199	181	4
S6	180	90	9.002
S7	208	194	3
S8	163	82	9
S9	180	119	8

**Table 7 materials-15-05607-t007:** Signal-to-noise ratio for hardness of the composite (larger is better).

Level	Stirring Speed (rpm)	Stirring Time (min)	SiC (wt.%)	MoS_2_ (wt.%)
1	44.42	44.33	43.72	44.31
2	**45.30**	45.16	45.06	**45.58**
3	45.24	**45.48**	**46.19**	45.08
Delta	0.88	1.15	2.48	1.26
**Rank**	**4**	**3**	**1**	**2**

**Table 8 materials-15-05607-t008:** Signal-to-noise ratio tensile strength of the composite (larger is better).

Level	Stirring Speed (rpm)	Stirring Time (min)	SiC (wt.%)	MoS_2_ (wt.%)
1	41.46	41.22	38.29	41.38
2	41.53	41.90	41.37	42.35
3	41.87	41.75	45.20	41.13
Delta	0.40	0.67	6.90	1.23
**Rank**	**4**	**3**	**1**	**2**

**Table 9 materials-15-05607-t009:** Signal-to-noise ratio for wear rate of composite (smaller is better).

Level	Stirring Speed (rpm)	Stirring Time (min)	SiC (wt.%)	MoS_2_ (wt.%)
1	−12.301	−15.035	−19.391	−16.701
2	−15.564	−16.009	−16.842	−15.177
3	−15.563	−12.383	−7.195	−11.549
Delta	3.263	3.626	12.196	5.152
**Rank**	**4**	**3**	**1**	**2**

**Table 10 materials-15-05607-t010:** Experimental results and S/N ratio of hardness, tensile strength, and wear rate values.

Designation	Experimental Result			*S*/*N Ratio*		
	Hardness (HV)	Tensile Strength (MPa)	Wear Rate (×10^−9^ kg/m)	Hardness	Tensile Strength	Wear Rate
S1	123	75	10	41.8193	37.5012	−20.0000
S2	183	129	7	45.2443	42.2205	−16.9020
S3	204	171	1	46.2096	44.6721	0.0000
S4	174	105	6	44.8010	40.3906	−15.5630
S5	199	181	4	45.9989	45.1464	−12.0412
S6	180	90	9.002	45.1136	39.0617	−19.0868
S7	208	194	3	46.3738	45.7753	−9.5424
S8	163	82	9	44.2224	38.3196	−19.0849
S9	180	119	8	45.1199	41.5036	−18.0618

**Table 11 materials-15-05607-t011:** Normalized data of the experimental results.

Designation	Normalized *S*/*N Ratio*		
	Hardness	Tensile Strength	Wear Rate
S1	0	0	1
S2	0.752	0.571	0.8451
S3	0.964	0.866	0
S4	0.655	0.13	0.778
S5	0.918	0.923	0.602
S6	0.723	0.188	0.954
S7	1	1	0.477
S8	0.527	0.123	0.954
S9	0.725	0.483	0.903

**Table 12 materials-15-05607-t012:** Deviation sequence and GRC of responses with ∂ = 0.5.

Designation	Deviation Sequence (Δ)			Grey Relational Coefficient (GRC)		
	Hardness	Tensile Strength	Wear Rate	Hardness	Tensile Strength	Wear Rate
S1	1	1	0	0.333	0.333	1
S2	0.248	0.429	0.155	0.668	0.538	0.763
S3	0.036	0.134	1	0.933	0.788	0.333
S4	0.345	0.87	0.222	0.592	0.365	0.693
S5	0.082	0.077	0.398	0.859	0.866	0.557
S6	0.277	0.812	0.046	0.643	0.382	0.916
S7	0	0	0.523	1	1	0.489
S8	0.473	0.877	0.046	0.514	0.363	0.916
S9	0.275	0.517	0.097	0.645	0.492	0.838

**Table 13 materials-15-05607-t013:** GRG and their order.

Designation	GRC			GRG	Order
	Hardness	Tensile Strength	Wear Rate		
**S1**	**0.333**	0.333	1	0.555	8
S2	0.668	0.538	0.763	0.656	5
S3	0.933	0.788	0.333	0.685	3
S4	0.592	0.365	0.693	0.550	9
S5	0.859	0.866	0.557	0.760	2
S6	0.643	0.382	0.916	0.647	6
S7	1	1	0.489	0.930	1
S8	0.514	0.363	0.916	0.598	7
S9	0.645	0.492	0.838	0.658	4

**Table 14 materials-15-05607-t014:** Signal-to-noise ratio for GRG for larger is better.

Level	Stirring Speed (rpm)	Stirring Time (min)	SiC (wt.%)	MoS_2_ (wt.%)
1	−4.021	−3.646	−4.454	−3.711
2	−3.786	−3.504	−4.163	−2.691
3	−2.911	−3.568	−2.100	−4.315
Delta	1.110	0.142	2.354	1.624
**Rank**	**4**	**3**	**1**	**2**

## Data Availability

All data available in the article itself.

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
