# Peer review of "Investigation of Mechanical and Tribological Behaviors of Aluminum Based Hybrid Metal Matrix Composite and Multi-Objective Optimization"

_materials, 2022, doi:10.3390/ma15165607_

Round 1

Reviewer 1 Report

Manuscript materials-1766076 Investigation of Mechanical and Tribological Behaviors of Aluminium Based Hybrid Metal Matrix Composite and Multi-Objective Optimization; shows mechanical and wear properties characterisation of MMC aluminium based composites. Metal matrix composites have proven their advantage over conventional alloys in high-performance applications in the automotive and aerospace industries. The research subject is widely analysed in the literature, so the article's scientific level is at a medium level. The authors only focus on the description of the experimental results. The background and recent investigation should be summarised in the Introduction more clearly in terms of properties and production for such materials. More literature research about Al-based composites’ behaviour in different working conditions is necessary here. In addition, the novelty of this paper should be confirmed based on the literature research. Also, deep and meaningful discussions are deficient in this paper. In addition, the manuscript should flow logically, and I would like to suggest that it should be carefully revised by an older scientist experienced in the field. The experimental results should be better organised, and some essential discussion is necessary. Some problems are not clearly presented, i.e. as mentioned by the authors, this paper also aimed to study the microstructure of MMC material after the stir casting process. In my opinion, the paper has not had enough information on microstructure (matrix + particles) and the mechanism of their interaction and effect on the material strength and wear should be emphasized here. In addition, the English language has serious flaws and needs to be deeply revised. Poor writing makes the paper's logical progression very hard to follow. Finally, the scientific content is not self-conclusive, and I miss a serious comparison against the results of literature on the same processes and materials.  Some of the major comments I have highlighted below:

1. English needs intensive revision and correction

2. Introduction is fair too short and is not closely related to the topic. I propose extending the Introduction section and more deeply explaining the MMC composites production, properties (advantages and disadvantages) and structure. Some more state of the art would be also precious here.

3. Section 2.4. could be transferred to Section 3 with some more microstructural analysis and images (i.e. SEM / SEM-EDS observation).

4. It would be good to extend section 3. Results and Discussion. Try not to start the section with the table. The short introduction would be valuable.

5. Please write the units correctly and add the spaces between

6. The macro figures of the samples are not clearly visible and hard to read without scale markers

7. The Figure captions should be rewritten

8. HV measurements load should be clearly indicated

9. The conclusions in the article are too general. I suggest re-editing the conclusions and extending the description of the impact of the realised research on the field of MMC materials.

There are no more comments that I felt to highlight. In general, the article requires comprehensive methodological and editorial refinement and proofreading. The issues presented in the article are important and suitable for publication in the Journal Materials, but to increase their scientific value, the article requires corrections and should be again widely analyzed by the Authors. I recommend the paper for publication in Journal Materials after major revision and performing another full review process.

Author Response

Title: Investigation of Mechanical and Tribological Behaviors of Aluminium Based Hybrid Metal Matrix Composite and Multi-Objective Optimization

Materials-ID-1766076

The author is very grateful to the reviewers for their constructive comments that helped improve the quality of the manuscript. The author made a serious effort to revise the manuscript according to the reviewer's comments. Details of the revisions made in response to the comments are summarized in the table below. Manuscript revisions are shown in yellow colored text.

REVIEWER-COMMENT-1

REVIEWER-COMMENT-1

AUTHOR RESPONSE

1# English needs intensive revision and correction

Reply: The authors are grateful for the valuable comment and suggestions, as per your comments, we have revised and corrected English.

2# Introduction is fair too short and is not closely related to the topic. I propose extending the introduction section and more deeply explaining the MMC composites production, properties (advantages and disadvantages) and structure. Some more state of the art would be also precious here.

Reply: The authors are grateful for the valuable comment and suggestions, as per your comments, we have extended the introduction part and correlated the introduction with the topics to improve the quality of the manuscript. .

3# Section 2.4. Could be transferred to Section 3 with some more microstructural analysis and images (i.e. SEM / SEM-EDS observation).

Reply: The authors are grateful for the valuable comment and suggestions, as per your comments, we have transferred the section and the microstructural analysis is removed and since images of optical microscope are used, the title is changed as identification of particle distribution in Al matrix.

4# It would be good to extend section 3. Results and Discussion. Try not to start the section with the table. The short introduction would be valuable.

Reply: The authors are grateful for the valuable comment and suggestions, as per your comments, section 3 are extended, short introduction has been given before the table in starting of the section.

5# Please write the units correctly and add the spaces between

Reply: The authors are grateful for the valuable comment and suggestions, as per your comments, the units are corrected and space is added in between.

6# The macro figures of the samples are not clearly visible and hard to read without scale markers

Reply: The authors are grateful for the valuable comment and suggestions, as per your comments, we have increased the quality and size of macro figures of samples to make it visible and readable to improve the quality of the manuscript.

7# The Figure captions should be rewritten

Reply: The authors are grateful for the valuable comment and suggestions, as per your comments, we have rewritten the figure caption.

8# HV measurements load should be clearly indicated

Reply: The authors are grateful for the valuable comment and suggestions, as per your comments, we have clearly indicated the HV measurement.

9# The conclusions in the article are too general. I suggest re-editing the conclusions and extending the description of the impact of the realised research on the field of MMC materials.

Reply: The authors are grateful for the valuable comment and suggestions, as per your comments, we have re-edited the conclusion and extended the description of the research on MMC materials.

Reviewer 2 Report

In this paper, the aluminium based MMC at varying stirring speed stirring time, and weight % of Silicon carbide powder has been tested and investigated.

From the analysis of the presented manuscript, I found the following remarks and questions:

1.      No information is given on the metallurgical state of the tested aluminum alloy, i.e. annealed or hardened state.

2.      Two equations (1) (2) have been used Normalized value Zij, what is the difference between them, please comment on this

3.      The equations (1) (2) (3) have been used but where those equations have been used ?, no results were described

4.      In my opinion, a schematic layout of the specimen with dimensions used in tensile and the hardness should be presented.

5.      There is no information on methodology for determination of fracture parameters values.

6.      The curve of the tensile test must be presented (True stress VS True strain) in my opinion with different strain rates .

7.      The comments in figures 8 and 9 must be detailed, could you add more comments ?  

Every experimental result should have a corresponding explanation. For example, the strain rate sensitivity of the provided materials in this work. However, written English should be carefully revised.

Author Response

Title: Investigation of Mechanical and Tribological Behaviors of Aluminium Based Hybrid Metal Matrix Composite and Multi-Objective Optimization

Materials-ID-1766076

The author is very grateful to the reviewers for their constructive comments that helped improve the quality of the manuscript. The author made a serious effort to revise the manuscript according to the reviewer's comments. Details of the revisions made in response to the comments are summarized in the table below. Manuscript revisions are shown in yellow colored text.

REVIEWER-COMMENT-2

REVIEWER-COMMENT-2

AUTHOR RESPONSE

1# No information is given on the metallurgical state of the tested aluminum alloy i.e. annealed or hardened state.

Reply: The authors are grateful for the valuable comment and suggestions. As per your comments, the enhancement of hardness of Al matrix was achieved by addition of reinforcement not with heat treatment.

2# Two equations (1) (2) have been used Normalized value Zij, what is the difference between them, please comment on this.

Reply: We sincerely appreciate the reviewer for the valuable comments and suggestions, as per your comments, we have identified the equation that equation (1) used for larger the better response (hardness and tensile strength), equation (2) used for smaller the better (wear rate) in normalization of S/N values.

3# The equations (1) (2) (3) have been used but where those equations have been used? No results were described.

Reply: We sincerely appreciate the reviewer for the valuable comments and suggestions, as per your comments, we have identified the equation that equation (1) used for larger the better response (hardness and tensile strength), equation (2) used for smaller the better (wear rate) in normalization of S/N values. The S/N value of hardness, tensile strength and wear rate on table10 have been normalized using this equation and shown on table 11. Eq (3) are used to determine the grey relational coefficient tabulated on table 12

4# In my opinion, a schematic layout of the specimen with dimensions used in tensile and the hardness should be presented.

Reply: The authors thank you for the valuable comment and suggestions, as per your comments, we have presented the schematic layout of the sample used for hardness, tensile and wear test.

5# There is no information on methodology for determination of fracture parameters values.

Reply: The author’s thank you for the valuable comment and suggestions, the fracture parameters values was not studied and included in this paper.

6# The curve of the tensile test must be presented (True stress VS True strain) in my opinion with different strain rates 

Reply: We sincerely appreciate the reviewer for the valuable comments and suggestions, but the aim of the study was to study the effect of process parameters on experimental numeric value of tensile strength. Therefore, that is why (True stress VS True strain) is not included in the paper.

7# The comments in figures 8 and 9 must be detailed, could you add more comments?  

Reply: We sincerely appreciate the reviewer for the valuable comments and suggestions, as per your comments, we have added more comments to describe the figures in detail.

Reviewer 3 Report

The introduction should provide more background concerning the use Taguchi Signal-to-noise method as well as the method itself. Relevant references about this method would improve the paper.

One important experimental data is missing: particle size of SiC particles.

The presentation of the results can be improved. This applies to the following items:

-       -   missing optical micrographs showing the distribution of the particles (the statement of homogeneous distribution in the conclusions is not proved by the single Figure 2)

-          -The head of table 4 is not clear.

-         - Numbers of experimental results should be rounded especially in the text - moreover all decimal digits are of no concern.

-          I would recommend to separate results and discussion. The results in a separate section would be more clear and the discussion should not be repeated so much.

There is a mismatch between Figure numbers and corresponding indication in the text.

Author Response

Title: Investigation of Mechanical and Tribological Behaviors of Aluminium Based Hybrid Metal Matrix Composite and Multi-Objective Optimization

Materials-ID-1766076

The author is very grateful to the reviewers for their constructive comments that helped improve the quality of the manuscript. The author made a serious effort to revise the manuscript according to the reviewer's comments. Details of the revisions made in response to the comments are summarized in the table below. Manuscript revisions are shown in yellow colored text.

REVIEWER-COMMENT-3

REVIEWER-COMMENT-3

AUTHOR RESPONSE

1# The introduction should provide more background concerning the use Taguchi Signal-to-noise method as well as the method itself. Relevant references about this method would improve the paper.

Reply: The authors are grateful for the valuable comment and suggestions, as per your comments, we have revised and added Taguchi Signal-to-noise method, the relevant reference.

2# One important experimental data is missing: particle size of SiC particles.

Reply: We sincerely appreciate the reviewer for the valuable comments and suggestions, we have added the particle size of SiC particles which are 270 mesh size (53 micrometer)

3# missing optical micrographs showing the distribution of the particles (the statement of homogeneous distribution in the conclusions is not proved by the single Figure 2)

Reply: The authors are grateful for the valuable comment and suggestions, as per your comments, we have added figures for all sample to state the distribution of particles in the conclusions

4# The head of table 4 is not clear.

Reply: We sincerely appreciate the reviewer for the valuable comments and suggestions, as per your comments, we have corrected the head of the table and changed to the determination of volume fraction of grain particles in composite

5# Numbers of experimental results should be rounded especially in the text - moreover all decimal digits are of no concern.

Reply: We really thank you for your suggestion. As per your comments, we have rounded the decimal experimental result number.

6# I would recommend to separate results and discussion. The results in a separate section would be more clear and the discussion should not be repeated so much.

Reply: We would like to appreciate your fruitful comments and suggestion.  As per your comments, we have presented the result and discussion separately.

7# There is a mismatch between Figure numbers and corresponding indication in the text?  

Reply: We sincerely appreciate the reviewer for the valuable comments and suggestions, as per your comments, we have revised and corrected the correlation between figure number and corresponding indication in the text

Round 2

Reviewer 1 Report

The Authors have made corrections according to reviewer's comments and article in such form has better scientific quality. However, the article should be proofread  once again while there are still some editorial mistakes (i.e. 1823 HV, figure caption more precisely presenting the pictures). English also requires improvement and should be checked 1by professional translator. I recommend manuscript for publication in Journal Materials

Author Response

Title: Investigation of Mechanical and Tribological Behaviors of Aluminium Based Hybrid Metal Matrix Composite and Multi-Objective Optimization

Materials-ID-1766076

The author is very grateful to the reviewers for their constructive comments that helped improve the quality of the manuscript. The author made a serious effort to revise the manuscript according to the reviewer's comments. Details of the revisions made in response to the comments are summarized in the table below. Manuscript revisions are shown in yellow colored text

REVIEWER-COMMENT-1

AUTHOR RESPONSE

#1: The Authors have made corrections according to reviewer's comments and article in such form has better scientific quality. However, the article should be proofread  once again while there are still some editorial mistakes (i.e. 1823 HV, figure caption more precisely presenting the pictures). English also requires improvement and should be checked 1by professional translator. I recommend manuscript for publication in Journal Materials

Reply: The authors are grateful for the valuable comment and suggestions, as per your comments, we have edited, revised and also corrected English.

Reviewer 2 Report

Thanks to the authors for the explanations and comments, all the recommendations and comments for the reviewer have been added to the main manuscript.

Author Response

Title: Investigation of Mechanical and Tribological Behaviors of Aluminium Based Hybrid Metal Matrix Composite and Multi-Objective Optimization

Materials-ID-1766076

The author is very grateful to the reviewers for their constructive comments that helped improve the quality of the manuscript. The author made a serious effort to revise the manuscript according to the reviewer's comments. Details of the revisions made in response to the comments are summarized in the table below. Manuscript revisions are shown in yellow colored text

REVIEWER-2-COMMENT

AUTHOR RESPONSE

#1: Thanks to the authors for the explanations and comments, all the recommendations and comments for the reviewer have been added to the main manuscript.

Reply: The authors are grateful for the valuable comment and suggestions, as per your comments, we have edited, revised and also corrected English.
